# A Systematic Review and Meta-Analysis on the Presence of *Escherichia coli* O157:H7 in Africa from a One Health Perspective

**DOI:** 10.3390/microorganisms13040902

**Published:** 2025-04-14

**Authors:** Namwin Siourimè Somda, Tomiwa Olumide Adesoji, Patience B. Tetteh-Quarcoo, Eric S. Donkor

**Affiliations:** 1Department of Medical Microbiology, University of Ghana Medical School, Korle Bu, Accra P.O. Box KB 4236, Ghana; namwin.somda@cnrst.gov.bf (N.S.S.); pbtetteh-quarcoo@ug.edu.gh (P.B.T.-Q.); 2Centre National de la Recherche Scientifique et Technologique (CNRST)/IRSAT/Département Technologie Alimentaire (DTA)/Bobo-Dioulasso, Bobo-Dioulasso 03 BP 2393, Burkina Faso; 3Department of Microbiology, Obafemi Awolowo University, Ile-Ife PC 22005, Nigeria; tom_adesoji@yahoo.com

**Keywords:** *E. coli* O157:H7, human, environment, animal, pooled prevalence, antimicrobial resistance, random-effects model

## Abstract

This systematic review aimed to assess the prevalence of *Escherichia coli* O157:H7 using a One Health approach, integrating data from human, animal, and environmental sources across Africa. Following PRISMA guidelines, studies reporting on *E. coli* O157:H7 in human, animal, and environment samples from African countries were retrieved from PubMed, Scopus, Web of Science, and Google Scholar. All data were analyzed using a binary random-effects model by the DerSimonian–Laird method at a 95% confidence interval. Out of 1757 publications generated, 56 from 9 countries including Ethiopia (17/56), South Africa (13/56), Nigeria (10/56), Egypt (9/56), Ghana (2/56), Tanzania (2/56), Benin (1/56), Namibia (1/56), and Senegal (1/56) were included. The pooled prevalence of *E. coli* O157:H7 was 4.7%, with the highest prevalence observed among animal samples (5.4%) followed by the environmental and human samples (3.4 and 2.8%, respectively). The pooled prevalence of antibiotic resistance was observed to be 96.5%, 82.8%, 76.8%, 70.7%, 62.1%, 50.4%, and 40.2% for cefoxitin, ampicillin, cefuroxime, nitrofurantoin, amikacin, amoxiclav, and ciprofloxacin, respectively. This distribution highlights the interconnectedness between animals, the environment, and human populations in the transmission and persistence of this pathogen and the need to implement a suitable and appropriate One Health pathogenic and antimicrobial resistance surveillance system in the African region.

## 1. Introduction

*Escherichia coli* (*E. coli*) is one of the many bacteria that live in the intestines of healthy humans and most warm-blooded animals. Most strains are harmless and help with digestion, but some strains can cause severe illness. *E. coli* O157:H7 serotype is a recognized food and waterborne pathogen that causes severe intestinal infection in humans [1,2]. It is the most common strain that causes acute hemorrhagic diarrhea, which may progress to hemolytic uremic syndrome, with systemic complications occurring more frequently in children [3]. *E. coli* O157:H7 can be differentiated from other *E. coli* by the production of a potent toxin, the Shiga toxin, which damages the lining of the intestinal wall leading to bloody diarrhea [4]. In Africa, Shiga toxin-producing *E. coli* (STEC) infections are estimated at 10,200 cases annually, with an incidence rate of 1.4 cases per 100,000 person per year [5]. Among these, *E. coli* O157:H7, a key STEC serotype, contributes 10% of reported infections [5,6].

Transmission occurs via the fecal–oral route after the consumption of contaminated, undercooked liquids, and foods or through person-to-person via fecal shedding [1]. Studies have identified cattle and other ruminants as the main reservoirs of *E. coli* O157:H7, with the pathogen being isolated from animals, food products, clinical samples, and environmental sources across all continents [5]. 

Reports have indicated prevalence of *E. coli* O157:H7 in cattle, sheep, goats, beef, meat products, chicken, dairy products, milk, fruits, and vegetables from several countries including Egypt, Algeria, and Libya [7]. In Ethiopia, the prevalence of *E. coli* O157:H7 was found to be 4% (95% CI = 3–5%) in foods of animal origin [8]. Similarly, in the North West Province of South Africa, the prevalence of *E. coli* O157:H7 was 9.5% from human, cattle, and pig samples [9]. According to community-based prevalence studies, the *E. coli* O157:H7 strain is responsible for 20 and 15.3% of *E. coli* infections in Nigeria [10] and Ethiopia [11], respectively. 

Recently, several systematic reviews in Africa have focused on the epidemiology of *E. coli* using the One Health approach [12,13,14,15]. However, to the best of our knowledge, there is limited pooled data on the prevalence of serotype *E. coli* O157:H7 in any systematic review using the One Health approach. Nevertheless, the few data reported have been those described in only a few African countries [16,17,18,19,20]. This systematic review was undertaken, to report on the pooled prevalence of *E. coli* O157:H7 using a One Health perspective across humans, animals, and the environment. Specifically, we aimed to (i) evaluate the pooled prevalence of *E. coli* O157:H7, (ii) identify the most favorable host to the spread of *E. coli* O157:H7 among these three components, and (iii) determine the pooled prevalence of the antibiotic resistance profile of the bacteria.

## 2. Materials and Methods

### 2.1. Study Design

This systematic review was conducted from 7 September 2024 to 28 October 2024 following the Preferred Reporting Items for Systematic Reviews and Meta-Analyses (PRISMA) guidelines (Appendix A). Studies focused on African countries and published between 1 January 2002 and 28 October 2024 were eligible for inclusion. Literature searches were conducted in PubMed, Scopus, Web of Science, and Google Scholar using Boolean operators to combine specific keywords such as *Escherichia coli* O157:H7, One Health, clinical and environmental samples, food products, and water sources, along with individual African country names (Appendix A). Only articles published in English and French were considered for inclusion. 

### 2.2. Study Eligibility Criteria

All search results from the different databases were exported to Excel software (v.13) and compiled. Rayyan was utilized to eliminate duplicates and to categorize and consolidate the results (available at https://www.rayyan.com/, accessed on 7 September 2024), while Endnote X9 software (v.20) was used to manage the collected publications and citations.

Inclusion criteria: Full-text research articles reporting on *E. coli* O157:H7 isolated in African countries if they investigated its presence in human–animal–environment, animal–human, human–environment, or animal–environment contexts. Studies were also required to provide details on the study population, sample source, number of isolates, and methods used to detect *E. coli* O157:H7. The literature search spanned all publications in the last 22 years (1 January 2002 to 28 October 2024) and was conducted by NSS and TOA.

Exclusion criteria: Articles were excluded if they contained incomplete information or focused solely on clinical, animal/livestock, or environmental samples without a broader context. Additionally, review articles, conference proceedings, duplicate publications, abstracts, posters, short communications, letters, and studies from non-African countries were not considered.

### 2.3. Screening and Data Extraction

Full texts of the selected publications were screened using Rayyan AI (https://www.rayyan.com/, accessed on 17 September 2024). The titles and abstracts relevant to the study question were carefully reviewed. Next, the selected papers at this stage further underwent full-text reviews, and only studies meeting the inclusion criteria were included in the review. Data were systematically collected and organized in an MS Excel spreadsheet by two reviewers, NSS and TOA. Both reviewers performed data extraction separately, and their results were compared for consistency. Extracted data included the study period, publication year, country, sampling population (specific animal, human, or environmental source/host), reservoirs studied (human–animal–environment, animal–human, animal–environment, human–environment), methods for *E. coli* O157:H7 detection, number of isolates from each source (animal, human, environment), prevalence of isolates, and multi-drug resistance pattern. 

### 2.4. Study Quality

The Joanna Briggs Institute (JBI) checklist for prevalence studies was used to assess the quality of the studies [21] and was carried out by two independent reviewers (NSS and TOA). The JBI checklist contains nine questions that were weighted as follows: 1 for a YES response or 0 for a NO response (Appendix A).

### 2.5. Data Analysis

The extracted data were used for descriptive statistics. Further analysis was carried out in multiple steps. Excel 2013 was used for data entry, while the meta-analysis, forest plots, and funnel plots of *E. coli* O157:H7 as well as the estimation of the country effect were conducted using the comprehensive meta-analysis. All data generated were analyzed using the binary random-effects model. This model is based on the DerSimonian–Laird method with a 95% confidence interval. The random-effects model was used to calculate the pooled prevalence of each sample source. The inverse variance index (I^2^) was used to quantify the heterogeneity across the studies and estimate the random-effects model. An I^2^ value of >75% indicated considerable heterogeneity. Statistical significance was denoted as a *p*-value < 0.05 [22]. Egger’s test was used to validate the asymmetry of the funnel plot.

## 3. Results

### 3.1. Characteristics of the Included Research Articles

A total of 1757 articles were generated in our initial search including 598 publications (34%) from Scopus, 543 (30.9%) from PubMed, 343 (19.5%) from Web of Science, and 271 (15.4%) from Google Scholar. After screening, a total of 56 (3.2%) articles were included in the review (Figure 1 and Table 1). 

Figure 2A shows the evolution of the number of publications according to year. Papers included in this review were found between 2004 and 2024. Out of the publications included in our study, 6/56 were published in 2014, 10/56 in 2017, 8/56 in 2020 and 2023, and 5/56 in 2024. Only a few articles were found in the other years, and the evolution of publications did not follow a normal curve. These 56 articles were found in only 9 countries including Ethiopia (17/56), South Africa (13/56), Nigeria (10/56), Egypt (9/56), Ghana (2/56), Tanzania (2/56), Benin (1/56), Namibia (1/56), and Senegal (1/56) (Figure 2B). Out of the 56 publications, 7 were excluded due to the absence of data on the sample population. Only 49 detailed information about the sample origin, sample number, isolate number, and methods. These 49 publications were used for the meta-analysis, of which 26 were from human–animal–environment samples, 12 from human–animal samples, 6 from human–environment samples, and 5 from animal–environment samples (Figure 2C).

### 3.2. Distribution of Samples, the Sampling Sources, and E. coli O157:H7 Identification Methods Used

Samples from human studies were obtained from stools (n = 19), urine (n = 1), and workers’ hand swabs (n = 7). Some studies did not specify the type of sample and mentioned only that there were human samples (n = 14). For the animal studies, most studies reported isolates from cattle (n = 16), meat and meat products (n = 15), beef (n = 14), pork (n = 8), and milk and dairy products (n = 7). A few studies reported on poultry (n = 4), chicken (n = 4), animal feces (n = 4), fish (n = 2), and mouton (n = 1). Some studies did not specify the type of samples and mentioned only that the samples were from animals (n = 3). Samples from the environment were obtained from water sources (n = 26), knives (n = 15), cutting boards (n = 6), manure or soil (n = 5), tables (n = 4), floors (n = 3), vegetables (n = 3), sinks (n = 3), vehicles for meat transport (n = 2), utensils (n = 2), door handles (n = 2), equipment (n = 2), walls (n = 2), bedrail (n = 1), cupboard (n = 1), stamp (n = 1), stretcher (n = 1), swivels (n = 1), and toilet seat (n = 1). Some studies did not specify the type of samples and only mentioned environmental samples (n = 3). The identification methods utilized in these studies included culture-based techniques (100%, n = 56), standard biochemical tests (91.1%, n = 51), PCR (53.6%, n = 30), serological (agglutination) test (16.1%, n = 9), serotyping using *E. coli* O157:H7 anti-sera (10.7%, n = 6), sequencing (8.9%, n = 5), pulse field gel electrophoresis (PFGE) (3.6%, n = 2), and immunochromatographic separation (1.8%, n = 1). In terms of antibiotic susceptibility testing, the Kirby–Bauer disk diffusion method was the only method used in all publications that described the antimicrobial susceptibility test (100%, n = 28).

### 3.3. Meta-Analysis

#### 3.3.1. Distribution of *E. coli* O157:H7 Prevalence by Country

In this review, the pooled prevalence of *E. coli* O157:H7 in Africa in the reviewed studies was 4.7% (95% CI: 3.4–6.5, I^2^ = 97.6%, *p* < 0.05) (Figure 3A). The highest prevalence was seen in animal samples with pooled prevalence of 5.4% (95% CI: 3.4–8.6, I^2^ = 96.9%, *p* < 0.05), followed by the environmental samples at 3.4% (95% CI: 2.3–5.0, I^2^ = 82.9%, *p* < 0.05), and human samples at 2.8% (95% CI: 1.8–4.2, I^2^ = 91.9%, *p* < 0.05). The presence of publication bias, represented by an asymmetrical funnel plot, was statistically confirmed with the random-effects model (*p* < 0.0001) (Figure 3B). The asymmetrical distribution of effect estimates, shown by a funnel plot of the study distribution, allowed us to examine the data according to the countries. Therefore, the pooled prevalence of *E. coli* O157:H7 was 12.6% (n = 13; 95% CI: 7.1–21.2; I^2^ = 98.1%; *p* < 0.05) in South Africa, 6.0% (n = 17, 95% CI: 5.5–6.5, I^2^ = 93.6%; *p* < 0.05) in Ethiopia, 4.8% (n = 10; 95% CI: 2.4–9.4; I^2^ = 96.2%; *p* < 0.05) in Nigeria, and 2.4% (n = 9; 95% CI: 0.8–6.8; I^2^ = 96.2%; *p* < 0.05) in Egypt. Notably, the prevalence of *E. coli* O157:H7 was 9.1% (n = 2) in Ghana, 8.8% (n = 1) in Benin, 4.1% (n = 1) in Namibia, 2.5% (n = 2) in Tanzania, and 1.7% (n = 1) in Senegal.

#### 3.3.2. Distribution of *E. coli* O157:H7 Prevalence According to Combination of Sample Sources

Combination human–animal–environment sources: Figure 4A shows the distribution of the pooled prevalence of *E. coli* O157:H7 from the 26 articles reviewed having human–animal–environment samples. According to the random effects analysis, this prevalence was 3.7% (95% CI: 2.0–6.8; I^2^ = 98.1%; *p* < 0.05). Figure 4B shows a funnel plot showing the distribution bias in effect estimates among studies examining the prevalence of *E. coli* O157:H7 in Africa. Among these 26 articles, the pooled prevalence of *E. coli* O157:H7 was 4.3% (95% CI: 1.3–13.4; I^2^ = 97.8%; *p* < 0.05) from animal samples (Figure 5A,B), followed by 3.1% (95% CI: 1.5–6.3; I^2^ = 66.4%; *p* < 0.05) from environment samples (Figure 6A,B), and 2.3% (95% CI: 1.0–4.9; I^2^ = 73.6%; *p* < 0.05) from human samples (Figure 7A,B). These publications were found in 6 countries from which the pooled prevalence of *E. coli* O157:H7 was 13.8% (n = 6; 95% CI: 4.4–35.8; I^2^ = 98.8%; *p* < 0.05) in South Africa, 4.1% (n = 5; 95% CI: 1.8–8.9; I^2^ = 93.3%; *p* < 0.05) in Nigeria, 2.8% (n = 11, 95% CI: 1.7–4.5; I^2^ = 96.2%; *p* < 0.05) in Ethiopia, 0.9% (n = 2; 95% CI: 0.5–1.8; I2 = 0%; *p* < 0.05) in Egypt, 4.1% (n = 1) in Namibia, and 0.4% (n = 1) in Tanzania.Combination of human–animal sources: In this review, 12 studies described concrete information on *E. coli* O157:H7 from human and animal sources. The average pooled prevalence was 4.0% (95% CI: 2.3–7.0; I^2^ = 96.9%; *p* < 0.05) (Figure 8A). The asymmetrical distribution of the effect estimates, shown by a funnel plot of the study distribution, allowed us to examine the data according to the countries and sample sources (Figure 8B). Specifically, the pooled prevalence of *E. coli* O157:H7 was 3.6% (95% CI: 1.8–6.9; I^2^ = 94.1%; *p* < 0.05) from the human samples and 6.7% (95% CI: 3.5–12.3; I^2^ = 95.6%; *p* < 0.05) from the animal samples. In this review, these publications were from 5 countries including Egypt (n = 5; 2.2% (95% CI: 0.6–8.1; I^2^ = 94.6%; *p* < 0.05), Ethiopia (n = 3; 7.1% (95% CI: 3.9–12.6; I^2^ = 93.9%; *p* < 0.05), Nigeria (n = 2; 4.8% (95% CI: 0.2–54.9; I^2^ = 98.7%; *p* < 0.05), Tanzania (n = 1; 9.8%), and Senegal (n = 1; 1.7%).Combination of human–environment sources: Our study highlighted 6 publications in this area, of which the pooled prevalence of *E. coli* O157:H7 was 6.0% (95% CI: 2.5–14.0; I^2^ = 97.2%; *p* < 0.05) (Figure 8C). The asymmetrical distribution of the effect estimates, which is shown by a funnel plot of the study distribution, led us to further examine the data according to countries and sample sources (Figure 8D). These research articles were described in 2 countries such as in South Africa (n = 4), where the pooled prevalence of *E. coli* O157:H7 was 10.9% (95% CI: 4.6–23.6; I^2^ = 97.3%; *p* < 0.05), and in Nigeria (n = 2), it was 1.9% (95% CI: 0.4–8.2; I^2^ = 60.6%; *p* > 0.05). The pooled prevalence of *E. coli* O157:H7 from the human samples was 2.7% (95% CI: 0.4–15.8; I^2^ = 94.3%; *p* < 0.05), and it was 3.7% (95% CI: 1.2–10.7; I^2^ = 94.2%; *p* < 0.05) from the environment samples.Combination of animal–environment sources: For this combination, the pooled prevalence of *E. coli* O157:H7 was 8.5% (95% CI: 6.5–11.0; I^2^ = 94.7%; *p* < 0.05) in the 5 publications reviewed (Figure 8E). Figure 8F is a funnel plot showing the distribution bias in the effect estimates among studies that examined the prevalence of *E. coli* O157:H7 in the animal–environment samples. Specifically, the pooled prevalence of *E. coli* O157:H7 from the animal samples in this combination was 5.7% (95% CI: 1.0–26.9; I^2^ = 94.1%; *p* < 0.05), and it was 3.6% (95% CI: 1.4–8.9; I^2^ = 69.9%; *p* < 0.05) in the environment samples. The prevalence of *E. coli* O157:H7 was 38% (n = 1) in Nigeria, 8.8% (n = 1) in Benin, 4.7% (n = 1) in Ethiopia, 2.2% (n = 1) in Egypt, and 0.02% (n = 1) in Ghana.

#### 3.3.3. Pooled Antimicrobial Resistance (AMR) Prevalence of *E. coli* O157:H7 Across the Human, Animal, and Environmental Studies in Africa

Out of the 56 studies included in this systematic review, only 28 studies described the antimicrobial resistance (AMR) patterns. Among these 28 publications, 23 presented the antibiotic resistance levels of *E. coli* O157:H7 to 15 antibiotics with all details, while the other 5 publications presented the antibiotic resistance level of *E. coli* generally. Out of twenty-three studies, fourteen were carried out on a combination of human–animal–environment sources, nine from human–animal sources, two from the human–environment area, and one from an animal-environment source. Considering the type of investigation (phenotypic or genotypic and phenotypic), all studies used the phenotypic investigation method, which mostly relies on the standard microbiological culture (Table 1). Only one publication from Nigeria used genotypic method to detect antimicrobial resistance gene; extended spectrum beta-lactamase (ESBL) genes were detected in *E. coli* O157:H7, indicating the presence of *bla_SHV_*, *bla_CTX-M_*, and *bla_TEM_* in one (12.5%), one (12.5%), and three (37.5%) isolates, respectively [38]. All the 23 publications reported the presence of antibiotic resistance in *E. coli* O157:H7. 

According to the random-effect analysis, the pooled prevalence was 77.4% for ampicillin (95% CI: 59.4–88.9; I^2^ = 75.0%; *p* < 0.05), 73.2% for tetracycline (95% CI: 55.3–85.7; I^2^ = 86.1; *p* < 0.05), 67.9% for trimethoprim/sulfamethoxazole (95% CI: 31.1–90.9; I^2^ = 82.4%; *p* = 0.342), 66.9% for gentamicin (95% CI: 56.6–75.8; I^2^ = 26.9%; *p* < 0.05), 63.5% for cefoxitin (95% CI: 6.4–97.8; I^2^ = 88.7%; *p* = 0.737), 53.4% for nitrofurantoin (95% CI: 37.8–68.4; I^2^ = 59.6%; *p* = 0.674), and 51.7% for cefuroxime (95% CI: 15.4–86.3; I^2^ = 71.9%; *p* = 0.941). However, the pooled prevalence was observed to be low in amikacin, 45.8% (95% CI: 26.9–65.9; I^2^ = 65.6%; *p* = 0.688), ceftazidime, 44.5% (95% CI: 9.3–86.3; I^2^ = 79.8%; *p* = 0.833), ceftriaxone, 43.5% (95% CI: 25.6–63.4; I^2^ = 70.0%; *p* = 0.528), amoxiclav, 43.1% (95% CI: 28.3–59.3; I^2^ = 81.3%; *p* = 0.408), ciprofloxacin, 35.9% (95% CI: 16.7–61.0; I^2^ = 91.6%; *p* = 0.269), nalidixic-acid, 25.5% (95% CI: 9.7–52.2; I^2^ = 0%; *p* = 0.07), and chloramphenicol, 23.9% (95% CI: 7.7–54.2; I^2^ = 93.3%; *p* = 0.09). There was a significant difference observed in all antibiotic-resistant *E. coli* O157:H7 described in this systematic review, except for ampicillin and tetracycline. This difference observed was not consistent; it sometimes showed a high prevalence in human studies, was sometimes high in animal studies, and at times high in environmental studies (Table 2). The observed antibiotic resistance in these studies was found to be the highest in animal studies compared to the human and environmental studies. The pooled antibiotic resistance prevalences were 96.5% for cefoxitin, 82.8% for ampicillin, 76.8% for cefuroxime, 70.7% for nitrofurantoin, 62.1% for amikacin, 50.4% for amoxiclav, and 40.2% for ciprofloxacin. Moreover, the pooled prevalence in trimethoprim/sulfamethoxazole was 82.9%, while it was 80.7% for tetracycline, and 48.7%, 45.7%, and 26.4% for ceftazidime, ceftriaxone, and chloramphenicol, respectively, in the human studies. Of note, imipenem was only described in human studies, of which the prevalence was 83.3%, while only gentamicin was reported in the environmental studies with a pooled prevalence of 72.6%. In the animal isolates, the resistance levels were higher in cefoxitin than in other antibiotics with 96.5% as the pooled prevalence. For human isolates, the pooled prevalence of *E. coli* O157:H7 was 83.3% to nalidixic-acid and imipenem.

### 3.4. Risk of Bias

In our review, the included studies reported high heterogeneity, as indicated by the I^2^ = 97.6% and Cochrane Q test (Q = 2317.28370, *p* < 0.0001). Visual inspection of the funnel plot showed a slight asymmetrical distribution. The intercept of the Eggers regression model was 1.33121 (95% CI: −11.01740–5.67955) with a t statistic of 6.27 and a *p*-value of 0.000. This finding suggests that potential publication bias in the included studies was unlikely (Figure 3B).

## 4. Discussion

This study was a systematic review of *E. coli* O157:H7 from the One Health approach, focusing simultaneously on animals, humans, and their environment. Our investigations showed that Ethiopia, South Africa, Nigeria, and Egypt were the African countries with the highest number of publications on *E. coli* O157:H7. Many factors could be attributed to this fact. According to Untaman et al. [74], the contribution of the number of publications in these countries could be due to the number of journal outlets, the high number of research institutions or universities, and the research specialization observed in these countries. Additionally, these countries are English-speaking nations, and the English language is most often used for scientific publications [75]. Recently, a systematic review and meta-analysis carried out on *E. coli* isolates from water in Africa showed that South Africa, Ethiopia, and Nigeria had the highest number of studies [14]. According to these authors, the higher socio-economic status of these countries in the region, and thus their ability to invest well in research and facilities, could explain this fact [14]. Moreover, not reporting data on *E. coli* O157:H7 from the One Health perspective in some African countries does not explain the absence of the bacterium, as there may be a high chance that this organism exists in these countries. Lupindua et al. showed that the lack of reports on *E. coli* O157:H7 isolation in some African countries could be due to the poor and insufficient diagnostic facilities even in some national reference laboratories, especially in rural settings where infections may be undiagnosed [5]. Most studies also used conventional culture methods for *E. coli* O157:H7 detection. This conventional method may be a limitation to the identification of *E. coli* O157:H7, since the evolution of molecular methods offers more robust and diverse methods for the detection of *E. coli* O157:H7 [76].

The data obtained from the research articles in this study did not present an exponential evolution in the publication years. In our investigation, most articles were published in 2017, 2020, and 2023. However, some systematic reviews in Africa have shown that the evolution of publications by year followed a normal curve [12,14,75,77].

To the best of our knowledge, this study is the first comprehensive systematic review and meta-analysis on the prevalence of *E. coli* O157:H7 simultaneously isolated from human, animal, and environment samples. This will contribute to the surveillance of this pathogenic bacteria and aid in designing preventive measures that will reduce the associated mortality of children under five years due to diarrhea in different regions of Africa. Over the last ten years, many studies have reported the incidence of *E. coli* O157:H7 in Africa [78,79,80,81,82]. However, data using the One Health perspective have been limited in the region.

It was observed that the pooled prevalence of *E. coli* O157:H7 from articles using the One Health approach in Africa was 4.7%. The random effects analysis showed that the pooled prevalence of the bacteria was 3.7%, 4%, 6%, and 8.5% in the human–animal–environment, human–animal, human–environment, and animal–environment publications, respectively. These results highlight the possibility of this bacteria being able to spread in different interfaces (human, animal, and environment). This could be linked to many factors, notably human activities. In terms of the sample source of *E. coli* O157:H7, the animal samples had the highest pooled prevalence of 5.4%. In Brazil, the pooled prevalence of *E. coli* O157:H7 in bovine meat and meat products was 1% lower than the prevalence in our findings [83]. According to Assefa and Bihon (2018), the pooled prevalence of *E. coli* O157:H7 in foods of animal origin in Ethiopia was 4% [84]. In Tunisia, *E. coli* O157:H7 isolated from cattle had a prevalence of 4.2% [85]. These results are slightly similar to the results of our findings. According to some reports, cattle are the primary reservoirs of *E. coli* O157:H7, and the consumption of beef and beef products have been identified as major sources of foodborne transmission [55]. Cattle could also be considered as the origin for *E. coli* O157:H7 spreading in environment, as they shed the pathogen normally through their feces. There is a high chance that these pathogens could contaminate vegetables, as farmers often use cow dung as natural fertilizers or irrigate vegetables with water that has been contaminated with cattle feces [7]. Our investigation showed that there was a variation in the prevalence of *E. coli* O157:H7 in foods and animal products from one region to another. This could be the result of geographical variations in slaughter hall conditions and handling practices (transportation trucks, carcass, and carcass with contact surfaces) [86].

From the environmental samples, our systematic review report the pooled prevalence of *E. coli* O157:H7 at 3.4%. Some environmental-based studies have reported near approximate prevalence; this is the case in Ethiopia and the United Kingdom, where the overall prevalence of *E. coli* O157:H7 was reported as 4.7% and 4.3%, respectively [87]. This bacterium is often considered as a pathogen frequently isolated from waters and wastewater. In African countries, vegetables are rinsed using various water sources including rivers and ponds close to the garden or selling site, increasing the risk of contamination with *E. coli* O157:H7 and other pathogens [88].

Our investigation showed that the pooled prevalence of *E. coli* O157:H7 from human samples was 2.8%. Generally, *E. coli* O157:H7 is most commonly implicated in human infections, especially food poisoning. Its transmission to humans mainly occurs through the consumption of contaminated foods such as raw or undercooked ground meat, raw milk, raw vegetables, and contaminated sprouted seeds [89]. The low prevalence of *E. coli* 0157:H7 reported in this review from humans could be due to the fact that many countries now observe an amelioration of sanitation, and the populace have started adopting better hygienic measures such as practices linked to food preparation and sanitation conditions.

In addition, the systematic review focused on 28 studies that highlighted the antibiotic resistance in *E. coli* 0157:H7 isolates. The pooled prevalence of antibiotic resistance in *E. coli* O157:H7 varied from one antibiotic to another and the different sample sources. For most of the antibiotics tested, resistance was observed more in the isolates from animal samples than from the environmental and human samples. Numerous studies that have reported antibiotic resistance in *E. coli* using the One Health approach have described similar results to our findings. This has been notably reported in Vietnam [90], Africa [14,91,92], and all over the world [93]. This fact is not surprising as the National Agency for Sanitary Safety of Food, the Environment, and Work (ANSES) has reported that farmers in many developing countries freely give antibiotics to their animals as soon as they are sick, without first checking whether they are indeed suffering from bacterial infections. These are often even used routinely on healthy animals to prevent infections or as growth promoters [94]. Therefore, animal and environmental samples, especially wastewater, are considered as potential sources of antibiotic resistance genes and multidrug resistance bacteria [88]. Moreover, the higher antibiotic resistance observed from animal and environmental samples calls for a lot of sensitization regarding the prudent use of antibiotics in these sectors, knowing that antibiotic-resistant bacteria reach humans indirectly through the food chain [95]. This could be through the consumption of contaminated food or food-derived products [95]. Therefore, there is a need for continuous and strong surveillance of bacterial infections from these three compartments (human, environment, and animal). Furthermore, infection reporting, technical staff training, the acquisition of laboratory equipment, implementation of common standard operating procedures, and the sharing of AMR data and expertise are needed in Africa to guide the required new approaches for the control and treatment of bacterial infections.

Only one publication in this systematic review used the genotypic method to detect antimicrobial resistance genes, out of which ESBL genes were detected in *E. coli* O157:H7. Therefore, it is recommended that future research should investigate the presence of genes associated with antibiotic resistance in *E. coli* 0157:H7 isolated from the One Health perspective. The results will provide a useful database on the molecular epidemiology of antibiotic-resistant *E. coli* O157:H7 in the context of One Health in Africa. This will help policymakers and scientists to develop ways to combat the spread of antibiotic resistance using the One Health approach.

There were, however, a number of limitations in this systematic review that should be considered. For instance, our study protocol was not registered on the standard PROSPERO platform like other studies. In addition, the available data were not representative and varied from one country to another. Furthermore, many of the reviewed articles had small sample sizes, potentially creating a bias in our statistical analyses.

## 5. Conclusions

In this systematic review, most of the *E. coli* 0157:H7 isolates were recovered from animals, followed by environmental and human samples, with antibiotic resistance being more common in animal-derived isolates. The systematic review emphasizes the interconnectedness between animals, the environment, and human populations in the transmission and persistence of this pathogen and the need to implement a suitable and appropriate One Health pathogenic and antimicrobial resistance surveillance system in the African region. A more coordinated approach, including standardized procedures, improved laboratory capacity, and better data sharing, will strengthen efforts to monitor pathogen spread and antimicrobial resistance under a One Health framework.

Our systematic review revealed that the available data on *E. coli* O157:H7 from a One Health approach in Africa are not representative and vary from one country to another.

## Figures and Tables

**Figure 1 microorganisms-13-00902-f001:**
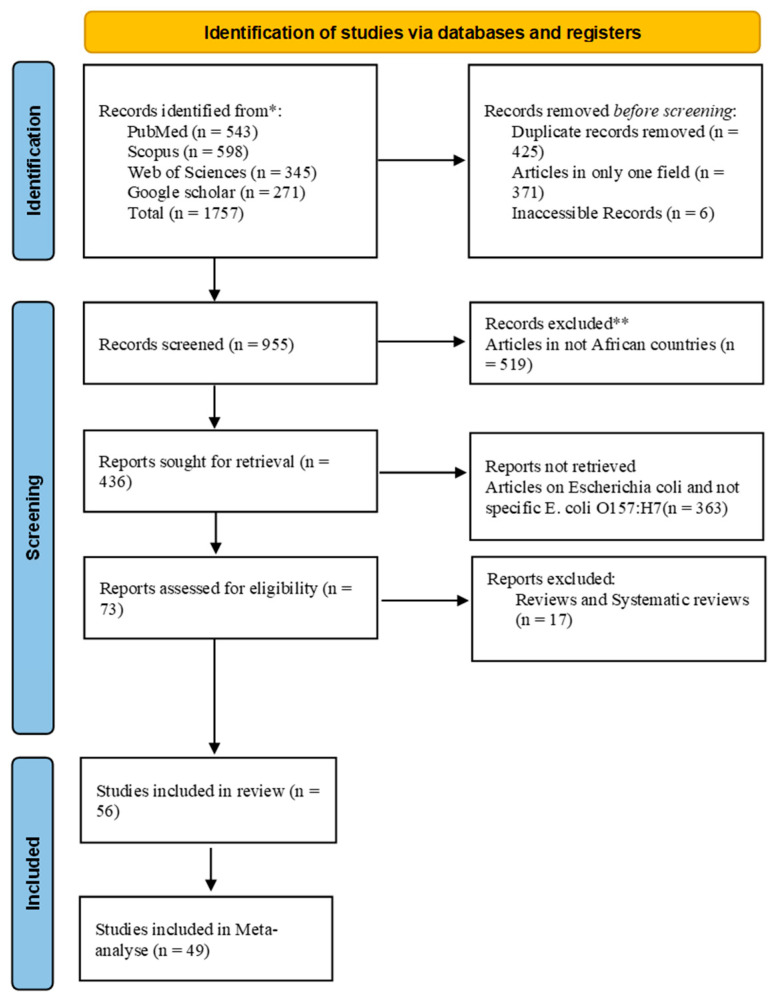
Flowchart (PRISMA flow diagram) of the systematic literature search, identification, screening, and article selection. * first screening; ** second screening (deep screening).

**Figure 2 microorganisms-13-00902-f002:**
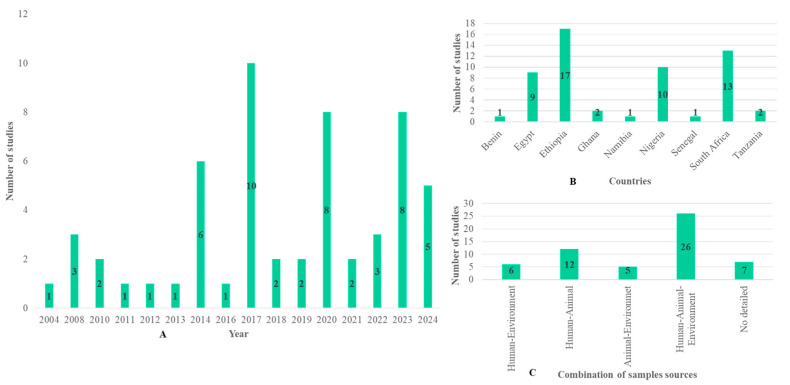
Summary of the selected studies showing the number of studies (**A**) per year, (**B**) per country, and (**C**) per combination of sample sources.

**Figure 3 microorganisms-13-00902-f003:**
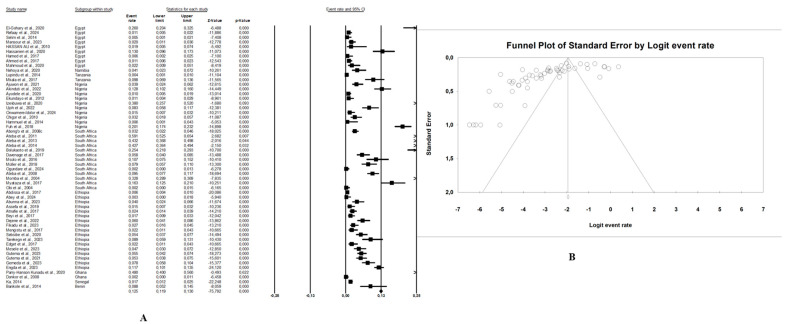
(**A**) Forest plot with the adjusted average prevalence of *E. coli* O157:H7 from the One Health perspective in Africa. Legend: Random effects mode (95% CI: 3.4–6.5, I^2^ = 97.6%, *p* < 0.05). X-axis is the proportion of countries reported in individual studies as listed along the Y-axis, with the range of proportion in the 95% confidence interval (CI). I^2^ = heterogenicity, *p* = *p*-value. The estimate of prevalence was calculated by pooling 56 studies using the random-effects model. (**B**) Funnel plot with the adjusted average prevalence of *E. coli* O157:H7 from the One Health perspective in Africa. Legend: The graph displays the standard error of the estimate (prevalence) on the Y-axis, while the X-axis represents the transformed proportions (prevalence), with individual studies represented by small circles. The 95% confidence interval is indicated by solid lines on the graph.

**Figure 4 microorganisms-13-00902-f004:**
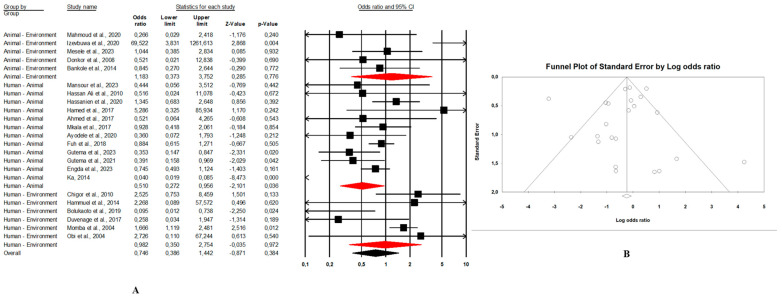
(**A**) Forest plot with the adjusted average prevalence of *E. coli* O157:H7 from human–animal–environment samples. (**B**) Funnel plot of *E. coli* O157:H7 from the human–animal–environment samples. Legend: Random effects mode (95% CI: 2.0–6.8; I^2^ = 98.1%; *p* < 0.05). X-axis is the proportion of *E. coli* O157:H7 reported in individual studies as listed along the Y-axis, with the range of proportion in the 95% confidence interval (CI). I^2^ = heterogenicity, *p* = *p*-value. The estimate of prevalence was calculated by pooling 26 studies that reported *E. coli* O157:H7 using the random-effects model.

**Figure 5 microorganisms-13-00902-f005:**
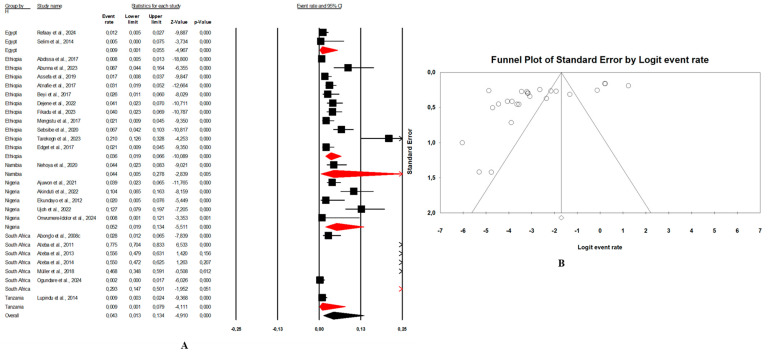
(**A**) Forest plot with the adjusted average prevalence of *E. coli* O157:H7 from animal samples. (**B**) Funnel plot of *E. coli* O157:H7 from animal samples. Legend: Random effects mode (95% CI: 1.3–13.4; I2 = 97.8%; *p* < 0.05). X-axis is the proportion of *E. coli* O157:H7 reported in individual studies as listed along the Y-axis, with the range of proportion in the 95% confidence interval (CI). I^2^ = heterogenicity, *p* = *p*-value. The estimate of prevalence was calculated by pooling all studies that reported *E. coli* O157:H7 from animal samples using the random-effects model.

**Figure 6 microorganisms-13-00902-f006:**
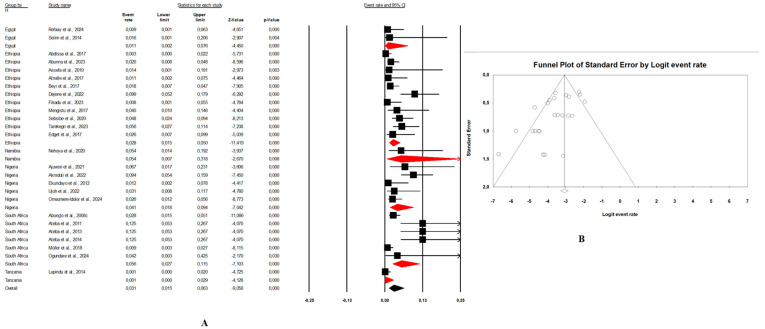
(**A**) Forest plot with the adjusted average prevalence of *E. coli* O157:H7 from environment samples. (**B**) Funnel plot of *E. coli* O157:H7 from environment samples. Legend: Random effects mode (95% CI: 1.5–6.3; I^2^ = 66.4%; *p* < 0.05). X-axis is the proportion of *E. coli* O157:H7 reported in individual studies as listed along the Y-axis, with the range of proportion in the 95% confidence interval (CI). I^2^ = heterogenicity, *p* = *p*-value. The estimate of prevalence was calculated by pooling all studies that reported *E. coli* O157:H7 from environment samples using the random-effects model.

**Figure 7 microorganisms-13-00902-f007:**
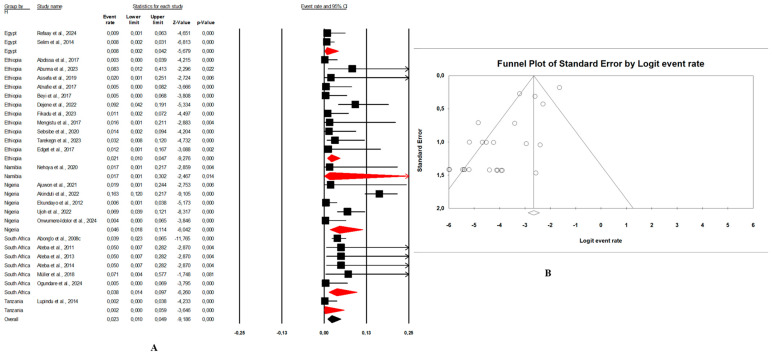
(**A**) Forest plot with the adjusted average prevalence of *E. coli* O157:H7 from human samples. (**B**) Funnel plot of *E. coli* O157:H7 from human samples. Legend: Random effects mode (95% CI: 1.0–4.9; I^2^ = 73.6%; *p* < 0.05). X-axis is the proportion of *E. coli* O157:H7 reported in individual studies as listed along the Y-axis, with the range of proportion in the 95% confidence interval (CI). I^2^ = heterogenicity, *p* = *p*-value. The estimate of prevalence was calculated by pooling all studies that reported *E. coli* O157:H7 from human samples using the random-effects model.

**Figure 8 microorganisms-13-00902-f008:**
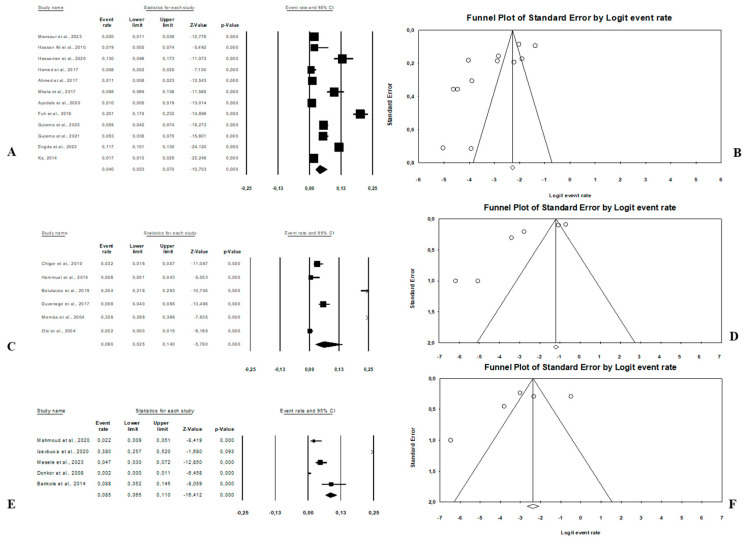
Forest plot and funnel plot with the adjusted average prevalence of *E. coli* O157:H7. (**A**,**B**) Human–animal samples; (**C**,**D**) human–environment samples; (**E**,**F**) animal–environment samples.

**Table 1 microorganisms-13-00902-t001:** Study characteristics.

Authors	Countries	Sources	Samples Numbers	Human Samples	Animal/Livestock Samples	Environment Samples	Number of Isolates	Clinical Isolates	Animal Isolates	Environment Isolates	Methods
[23]	Egypt	Bulk tank milk, milking utensils, knife swabs, wall swabs, workers’ hands swabs.	200	N/A	N/A	N/A	52	14	18	20	Culture, biochemical tests
[24]	Egypt	Chicken, beef, cutting board, and cutting knife, food handler (positive hand swabs)	648	108	432	108	7	1	5	1	Culture, biochemical tests, PCR, antisera anti-*E. coli* O157
[25]	Egypt	Fresh water, human stools and urine, animal stools, meat products, poultry products, seafood, dairy products	384	254	99	31	2	2	0	0	Culture, biochemical tests, PCR, genomics
[19]	Egypt	Chicken paneer, chicken burger, chicken luncheon, minced meat, beef burger, and Kariesh cheese, stools	550	100	450	0	11	1	10	0	Culture, biochemical tests, serotyping, PCR
[26]	Egypt	Minced beef, chicken fillet, chicken legs, children stools.	103	28	75	0	2	0	2	0	Culture, biochemical tests, serotyping
[27]	Egypt	Milk products and stools samples	300	150	150	0	39	22	17	0	culture, biochemical tests, serotyping
[28]	Egypt	Meat, luncheon, sausage, and beef burger, Karish cheese samples, and human stool	310	50	260	0	2	1	1	0	Culture, biochemical tests, PCR
[29]	Egypt	Meat samples, human stool samples	700	150	550	0	8	1	7	0	Culture, biochemical tests, ERIC-PCR
[30]	Egypt	Cattle carcasses, floors, doors, walls, knives, swivels, stamps, water gourds	230	0	110	120	5	0	1	4	Culture, biochemical tests, PCR
[20]	Namibia	Meat, equipment, hand swabs	270	29	204	37	11	0	9	2	Culture, biochemical tests, PCR
[31]	Tanzania	Human stool, soil, water, andfecal samples from cattle	1046	200	446	400	4	0	4	0	Culture, biochemical tests, PCR
[32]	Tanzania	Cattle and humans	307	107	200	0	30	10	20	0	Culture, biochemical tests, PCR
[33]	Nigeria	Cow carcass swabs, cecal content samples, water samples, hand swabs, and knife swabs	415	25	360	30	16	0	14	2	Culture, biochemical tests, latex kit
[17]	Nigeria	Stool, water samples, meat, skin and visceral organs from sheep, cattle, goat, and poultry	508	227	154	127	65	37	16	12	Culture, biochemical tests, PCR
[34]	Nigeria	Raw meat, fish, samples retailed, butchers’ processing tables, and utensils used in retailing meat and fish	823	394	429	0	8	2	6	0	Culture, biochemical tests, PCR
[35]	Nigeria	Stool specimens from undergraduates, food vendors, and specimens from non-human sources	366	180	100	86	4	1	2	1	Culture, biochemical tests, latex agglutination
[36]	Nigeria	Beef samples, processing water samples, table swabs and entrails samples	50	0	30	20	19	0	19	0	Culture, biochemical tests, serological test
[37]	Nigeria	Abattoir meat, abattoir waste water, roadside butchers’ meat, patients’ stool sample, food sellers’ stool	349	159	126	64	29	11	16	2	Culture, biochemical tests, serotyping
[38]	Nigeria	Slaughter floor, meat hooks, butchers’ hands, butchers’ knives, meat sellers’ table, meat sellers’ hands, meat sellers’ knives	406	116	58	232	6	0	0	6	Culture, biochemical tests
[39]	Nigeria	Diarrheal stools and surface waters	340	112	0	228	11	6	0	5	Culture, biochemical tests, agglutination test
[40]	Nigeria	Nurses’ hand swab, nurses’ table top, door knob/handle, toilet seat, operation table, sink, stretcher, floor, bedrail, and cupboard	160	20	0	140	1	0	0	1	Culture, biochemical tests, latex agglutination
[41]	Nigeria	Children stool and raw bovine meat	726	366	360	0	146	70	76	0	Culture, biochemical tests, latex agglutination
[42]	South Africa	Drinking water, meat and vegetables, andstools	900	360	180	360	29	14	5	10	Culture, biochemical tests, PCR
[43]	South Africa	Beef, pork, water, human, and animal species	220	20	160	40	130	1	124	5	Culture, biochemical tests, PCR
[44]	South Africa	Pigs, cattle, pork, beef, humans, and water samples	220	20	160	40	95	1	89	5	Culture, biochemical tests, REP, and ERIC PCR, ISR and BOXAIR PCR
[45]	South Africa	Cattle, pigs and humans, water samples	220	20	160	40	94	1	88	5	Culture, biochemical tests, ERIC PCR
[18]	South Africa	Run-off water, sewage water, surface water, wastewater, bloody, loose, mucoid, watery	520	250	0	270	132	1	0	11	Culture, biochemical tests, PCR, Sequencing, PFGE
[46]	South Africa	Water, fruit, hands	428	57	0	371	25	1	0	24	Bacteriological culture, biochemical tests, PCR
[47]	South Africa	Raw milk, cattle udder, milking machines, and worker’s hand swabs	252	N/A	N/A	N/A	27	N/A	N/A	N/A	Culture, biochemical tests, PCR
[48]	South Africa	Water samples, cattle and pig fecal samples, human fecal samples	403	6	62	335	32	0	29	3	Culture, PCR immunomagnetic separation (IMS), sequencing
[49]	South Africa	Poultry, swine, human hand swabs, and abattoir/farms run-off water	537	108	418	11	1	0	1	0	Culture, biochemical tests, PCR, sequencing
[50]	South Africa	Cattle, pigs, and humans	800	N/A	N/A	0	76	3	73	0	Culture, biochemical tests, serotyping
[51]	South Africa	Water samples and stool swabs	540	360	0	180	177	131	0	46	Culture, biochemical tests, PCR
[52]	South Africa	Drinking water, dairy wastewater and irrigation water, rectal samples from cattle	288	0	180	108	47	0	N/A	N/A	Culture, biochemical tests, PCR
[53]	South Africa	River samples were collected, and diarrheic stool samples	480	252	0	228	1	1	0	0	Culture, biochemical tests, PCR
[54]	Ethiopia	Fecal sample, skin swab, intestinal mucosal swab, carcass internal swab, carcass external swab, environmental swabs, carcass, hands, cutting board, knife, stool	2482	195	1975	312	16	0	15	1	Culture, biochemical tests, PCR
[16]	Ethiopia	Carcass surface, abattoir worker’s hand, knives, carcass wash water, cattle feces, abattoir effluent	384	30	280	74	1			1	Culture, biochemical tests, PCR
[55]	Ethiopia	Fecal, carcass swab, knife swabs, hand swabs, water/wastewater	352	12	92	248	14	1	8	5	Culture, biochemical tests, PCR
[56]	Ethiopia	Filleted fish swab, filleted fish muscle (tissue, whole fish (skin) swab, knife and cutting, board swab, ready to eat fish, workers’ hand swab, container swab	410	24	352	34	6	0	6	0	Culture, biochemical tests, PCR
[57]	Ethiopia	Fecal sample, carcass swab, knife swab, personnel hand swab, meat transporters cloth swab, meat sample, butcher men hand swab, cutting board swab, knife swab	630	90	450	90	15	0	14	1	Culture, biochemical tests
[58]	Ethiopia	Carcass swab, hand swab, knife swab, cutting board swab, minced beef	525	110	195	220	9	0	5	4	Culture, biochemical tests
[59]	Ethiopia	Milk, water, milker hand swab	450	65	294	91	27	6	12	9	Culture, biochemical tests, serological test
[60]	Ethiopia	Slaughtered cattle feces, carcass swabs, tap water, butcher hand, and knife swabs	516	93	300	123	14	1	12	1	Culture, biochemical tests, latex agglutination kit, PCR
[61]	Ethiopia	Raw beef meat, Environmental sample, Equipment, Workers hand, Contact surface, Balance, Vehicle, Cutting board and table Respondents	370	30	290	50	8	0	6	2	Culture, biochemical tests
[62]	Ethiopia	Meat sample, cecal content, hand swabs, knife swabs, protective cloth swabs, transport vehicles swab	502	70	270	165	27	1	18	8	Culture, biochemical tests, latex agglutination test
[63]	Ethiopia	Carcass, hand, knife, hook	248	62	62	124	22	2	13	7	Bacteriological culture, biochemical tests, andlatex agglutination tests
[64]	Ethiopia	Raw beef meat, environmental sample, equipment, workers’ hand, contact surface, balance, vehicle, cutting board, and table respondents	370	40	290	76	8	0	6	2	Bacteriological culture, biochemical tests
[65]	Ethiopia	Milk samples, feces samples, water, and manure samples	408	0	208	100	19	0	13	6	Bacteriological culture, biochemical tests
[66]	Ethiopia	Cattle, beef, and humans	793	216	507	0	44	6	38	0	Bacteriological culture, biochemical tests, whole-genome sequencing (WGS), MLST
[67]	Ethiopia	Cattle, beef, and humans	583	216	367	0	31	6	25	0	Bacteriological culture, biochemical tests, PFGE
[68]	Ethiopia	Samples included livestock fecal samples and soil samples	539	0	462	77	42	0	N/A	N/A	Culture, biochemical tests
[69]	Ethiopia	(Stools) diarrheic patients and cattle	1378	1149	229	0	161	128	33	0	Culture, latex agglutination test, PCR
[70]	Ghana	22 rinse water samples, 33 fecal matter samples, 33 bench top samples, and 60 freshly dressed chicken carcasses	148	0	60	88	71	0	N/A	N/A	Bacteriological culture, biochemical tests, PCR
[71]	Ghana	Vegetables, irrigation water, manure soil samples, stools of livestock	642	0	250	392	1	0	0	1	Bacteriological culture, biochemical tests
[72]	Senegal	Beef, pork, mouton, stools	1777	1667	110	0	31	13	18		Bacteriological culture, biochemical tests
[73]	Benin	Cattle, pigs, tables, knives, dropping of poultry, vegetable, irrigation water	148	0	74	74	13	0	6	7	Bacteriological culture, biochemical tests, serological test

Legend: N/A was used to designate non-applicability.

**Table 2 microorganisms-13-00902-t002:** Subgroup meta-analysis of the pooled antimicrobial resistance (AMR) prevalence of *E. coli* O157:H7 across the human, animal, and environment studies.

Antibiotics	Human Samples	Animal Samples	Environment Samples	Total
Ciprofloxacin				
Pooled prevalence %	25.1	40.2	33.3	35.9
95% CI; I^2^; *p*	7.7–57.4; I^2^ = 88.2; *p* = 0.123	17.9–67.5; I^2^ = 85.6; *p* = 0.492		16.7–61.0; I^2^ = 91.6; *p* = 0.269
Number of studies	3	4	1	5
Amoxiclav				
Pooled prevalence %	43.8	50.4	16.7	43.1
95% CI; I^2^; *p*	22.0–68.3; I^2^ = 87.6; *p* = 0.63	28.4–72.2; I^2^ = 65.5; *p* = 0.75		28.3–59.3; I^2^ = 81.3; *p* = 0.408
Number of studies	7	4	1	7
Ampicillin				
Pooled prevalence %	72.6	82.8	72.6	77.4
95% CI; I^2^; *p*	63.2–80.4; I^2^ = 0; *p* = 0	55.9–94.8; I^2^ = 79.9; *p* = 0.021	47.9–88.5; I^2^ = 27.8; *p* = 0.072	59.4–88.9; I^2^ = 75.0; *p* = 0.004
Number of studies	7	9	6	11
Gentamicin				
Pooled prevalence %	71.9	65.9	72.6	66.9
95% CI; I^2^; *p*	52.1–85.7; I^2^ = 0; *p* = 0.031	54.1–77.9; I^2^ = 44.8; *p* = 0.032	47.9–88.5; I^2^ = 27.8; *p* = 0.072	56.6–75.8; I^2^ = 26.9; *p* = 0.002
Number of studies	3	5	6	12
Imipenem				
Pooled prevalence %	83.3	N/A	N/A	83.3
95% CI; I^2^; *p*		N/A	N/A	
Number of studies	1	N/A	N/A	1
Ceftriaxone				
Pooled prevalence %	45.7	35.9	N/A	43.5
95% CI; I^2^; *p*	0.254–0.676; I^2^ = 60.2; *p* = 0.7	23.8–50.2; I^2^ = 30.9; *p* = 0.053	N/A	25.6–63.4; I^2^ = 70.0; *p* = 0.528
Number of study	3	2	N/A	3
Nitrofurantoin				
Pooled prevalence %	35.9	70.7	62.1	53.4
95% CI; I^2^; *p*	26.2–47.0; I^2^ = 14.2; *p* = 0.014	49.6–85.5; I^2^ = 50.5; *p* = 0.054	35.9–82.8; I^2^ = 0; *p* = 0.366	37.8–68.4; I^2^ = 59.6; *p* = 0.674
Number of studies	3	3	2	4
Tetracycline				
Pooled prevalence %	80.7	70.3	66.7	73.2
95% CI; I^2^; *p*	62.9–91.1; I^2^ = 79.8; *p* = 0.002	46.7–86.5; I^2^ = 79.0; *p* = 0.089	13.0–96.4; I^2^ = 66.2; *p* = 0.60	55.3–85.7; I^2^ = 86.1; *p* = 0.013
Number of studies	7	7	2	8
Ceftazidime				
Pooled prevalence %	48.7	40.3	47.4	44.5
95% CI; I^2^; *p*	33.6–64.0; I^2^ = 0; *p* = 0.873	3.7–92.2; I^2^ = 85.5; *p* = 0.788	6.5–92.1; I^2^ = 61.2; *p* = 0.935	9.3–86.3; I^2^ = 79.8; *p* = 0.833
Number of studies	2	3	2	4
Cefuroxime				
Pooled prevalence %	53.8	76.8	36.4	51.7
95% CI; I^2^; *p*	38.3–68.6; I^2^ = 0; *p* = 0.631	7.1–99.3; I^2^ = 84.5; *p* = 0.534	8.9–77.2; I^2^ = 58.9; *p* = 0.539	15.4–86.3; I^2^ = 71.9; *p* = 0.941
Number of studies	2	2	3	4
Trimethoprim/sulfamethoxazole				
Pooled prevalence %	82.9	67.9	66.7	67.9
95% CI; I^2^; *p*	53.7–95.3; I^2^ = 34.9; *p* = 0.215	12.0–97.1; I^2^ = 86.5; *p* = 0.592	13.0–96.4; I^2^ = 66.2; *p* = 0.6	31.1–90.9; I^2^ = 82.4; *p* = 0.342
Number of studies	2	3	2	5
Cefoxitin				
Pooled prevalence %	49.2	96.5	65.1	63.5
95% CI; I^2^; *p*	0.01–0.99; I^2^ = 88.6; *p* = 0.989	79.1–99.5; I^2^ = 0; *p* = 0.001	5.9–98.2; I^2^ = 80.0; *p* = 0.72	6.4–97.8; I^2^ = 88.7; *p* = 0.737
Number of studies	2	2	3	4
Nalidixic acid				
Pooled prevalence %	83.3	20.0	83.3	25.5
95% CI; I^2^; *p*	19.4–99.0; I^2^ = 0; *p* = 0.299	2.4–71.7; I^2^ = 46.2; *p* = 0.241	19.4–99.0; I^2^ = 0; *p* = 0.299	9.7–52.2; I^2^ = 0; *p* = 0.07
Number of studies	1	2	1	2
Amikacin				
Pooled prevalence %	23.8	62.1	N/A	45.8
95% CI; I^2^; *p*	8.1–52.5; I^2^ = 35.9; *p* = 0.071	22.9–90.0; I^2^ = 81.7; *p* = 0.57	N/A	26.9–65.9; I^2^ = 65.6; *p* = 0.688
Number of studies	2	2	N/A	2
Chloramphenicol				
Pooled prevalence %	26.4	16.2	N/A	23.9
95% CI; I^2^; *p*	8.4–58.4; I^2^ = 93.0; *p* = 0.141	1.2–75.1; I^2^ = 87.9; *p* = 0.241	N/A	7.7–54.2; I^2^ = 93.3; *p* = 0.09
Number of studies	3	3	N/A	4

Legend: N/A was used to designate non-applicable.

## Data Availability

All data generated or analyzed are included in this review.

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
