# Peer review of "A Systematic Review and Meta-Analysis on the Presence of Escherichia coli O157:H7 in Africa from a One Health Perspective"

_microorganisms, 2025, doi:10.3390/microorganisms13040902_

Round 1

Reviewer 1 Report

Comments and Suggestions for Authors

Dear Authors, congratulations from me for the excellent work of Systematic Review and Meta-Analysis.

The Title does not clarify the main focus of the work done. If the Authors agree, I would change it to: "A Systematic Review and Meta-analysis on the presence of Escherichia coli O157:H7 in Africa from a One Health perspective" or "The presence of Escherichia coli O157:H7 in Africa Countries from a One Health perspective: A Systematic Review and Meta-analysis"

Keywords: I would change them because they are too repetitive of what is already present in the title. I would add some words that better characterize the huge study carried out, e.g. "pooled data"

Page 1, Line 29: Escherichia coli, should always be reported in italics. In the manuscript, it often appears in a "normal" style.

Page 2, Lines 77-85: I would avoid writing all the countries researched, it becomes a list of little use. Perhaps it is better to indicate: (OR all the countries of the African continent).

Page 2, Line 88: Replace reviewed with "considered" otherwise it seems that the review work has been done for all 1757 results.

Page 3, line 108: The acronym that identifies one of the two researchers who evaluated the results of the research should be NSS and not SNS.

Page 19, lines 352-354: I believe that the reported reflection is not pertinent and does not justify a possible lower scientific production. I would eliminate it.

I believe that it is necessary to make the small corrections indicated for the manuscript to be ready for publication.

Author Response

Comments 1: The Title does not clarify the main focus of the work done. If the Authors agree, I would change it to: "A Systematic Review and Meta-analysis on the presence of Escherichia coli O157:H7 in Africa from a One Health perspective" or "The presence of Escherichia coli O157:H7 in Africa Countries from a One Health perspective: A Systematic Review and Meta-analysis"

Response 1: A Systematic Review and Meta-analysis on the presence of Escherichia coli O157:H7 in Africa from a One Health perspective. Thank you for pointing this out. We agree with this comment. Therefore, we have been made the change as noted in the revised manuscript page 1, Line 2.

Comments 2: Keywords: I would change them because they are too repetitive of what is already present in the title. I would add some words that better characterize the huge study carried out, e.g. "pooled data"

Response 2: We agree with this comment. Therefore, we have been made the change as noted in the revised manuscript page 1, Lines 27-28

Comments 3: Page 2, Lines 77-85: I would avoid writing all the countries researched, it becomes a list of little use. Perhaps it is better to indicate: (OR all the countries of the African continent).

Response 3: We agree with this comment. Therefore, we have been made the change as noted in the revised manuscript page 2, Line 78

Comments 4: Page 2, Line 88: Replace reviewed with "considered" otherwise it seems that the review work has been done for all 1757 results.

Response 4: We agree with this comment. Therefore, we have been made the change as noted in the revised manuscript page 2, Line 81

Comments 5: Page 3, line 108: The acronym that identifies one of the two researchers who evaluated the results of the research should be NSS and not SNS.

Response 5: We agree with this comment. Therefore, we have been made the change as noted in the revised manuscript page 2, Line 92, page 3, Lines 101, 113, page 21, Lines 459, 460, 461

Comments 6: Page 19, lines 352-354: I believe that the reported reflection is not pertinent and does not justify a possible lower scientific production. I would eliminate it.

Response 6: We agree with this comment. Therefore, we have been made the change as noted in the revised manuscript, Discussion section page 19, Line 345

Reviewer 2 Report

Comments and Suggestions for Authors

In general, the manuscript entitled “A Systematic Review and Meta-Analysis of Escherichia coli O157:H7 in Africa from a One Health Perspective” is a well-draft manuscript, but the manuscript needs some corrections and editions as follows:

1. Abstract

1.      Line 13. Replace “samples published in PubMed, Scopus, Web of Science, and Google Scholar databases were obtained using specific keywords.” With “samples, were retrieved from the PubMed, Scopus, Web of Science, and Google Scholar databases using specific keywords.”

2.      L 18-20. Replace “The pooled prevalence of E. coli O157:H7 was 4.7%. The pooled prevalence of E. coli O157:H7 from animal samples was 5.4% followed by isolates from environmental samples, 3.4% and 2.8% from isolates from human samples.” With “The pooled prevalence of E. coli O157:H7 was 4.7%, with the highest prevalence observed among animal samples (5.4%) followed by environmental, and human samples (3.4 and 2.8%, respectively).”

2. Introduction

3.      Line 29. Replace “Escherichia coli” WithEscherichia coli (E. coli)”

4.      L35. Replace “leading” With “leading to”

5.      L 38-39. Replace “In Africa, Shiga toxin-producing Escherichia coli (STEC) infections amount to approximately” With

“In Africa, the incidence of Shiga toxin-producing E. coli (STEC) infections is approximately”

6.      L 40-41. Replace “contributes 10% to this burden” With “accounts for 10% of this burden.”

7.      L 46. Replace “in; cattle, sheep, goats, beef and meat products, chicken, dairy products, milk, fruits and vegetables; from”. With “in cattle, sheep, goats, beef, meat products, chicken, dairy products, milk, fruits and vegetables from”

8.      L 51. “E. coli” The name of bacteria should be italic, apply this comment throughout the manuscript.

9.      L 51-52.  Replace “E. coli O157:H7 strain is responsible for 20% of E. coli infections in Nigeria [10], and 15.3% in Ethiopia [11], from community-based prevalence studies.” With

“According to the community-based prevalence studies, E. coli O157:H7 strain is responsible for 20, and 15.3% of E. coli infections in Nigeria [10], and Ethiopia [11], respectively.

10.  L 52-54. Replace “In recent years in Africa, several data reported in many systematic reviews have focused on the epidemiology of Escherichia coli using the One Health approach” With

“Recently, several systematic reviews in Africa have focused on the epidemiology of E. coli using the One Health approach”

11.  L 95. Replace “was undertaken and it is the first study that has reported the pooled prevalence” With “was undertaken, for the first time, to report the pooled prevalence”

2. Materials and Methods

12.  L 133. “p” Should be italic, APPLY THIS COMMENT THROUGHOUT THE MANUSCRIPT.

3. Results

13.  L 174, 182, 183, 185, 191. “E. coli”. The name of bacteria should be italic, APPLY THIS COMMENT THROUGHOUT THE MANUSCRIPT.

14.   157. “PFGE” Write the full name followed by abbreviation in brackets.

15.  139. Replace “(Fig.1) (Table 1).” With “(Fig.1 and Table 1).”

16.  L184-196. “p” Should be italic, apply this comment throughout the manuscript.

17.  L184. “AMR” Write the full name.

18.  L 286. Replace “antimicrobial resistance” With

“antimicrobial resistance (AMR)”

19.  L295. “ESBL” Write the full name followed by abbreviation in brackets.

20.  L296. “blaSHV, blaCTX-M, and blaTEM” The genes’ name should be italic, apply this comment throughout the manuscript.

21.  L 321. ReplaceIt is worthy of note that imipenem” With “Of note, in the current systematic review imipenem”

22.  L 323. Remove in this systematic review”

23.  L 304. Table 2. “AMR” Write the full name.

4. Discussion

24.  L 346-347. Replacewhere majority of the publications on E. coli O157:H7 isolated using the One Health perspective can be found” Withwith the highest number of publications on E. coli O157:H7 isolated through the One Health approach

25.  L 350. Replace “An other hand,” With “Additionally,”

26.  L383. Remove “respectively”

27.  L384. Replace “publications.” With “publications, respectively”

Author Response

Comments 1: Line 13. Replace “samples published in PubMed, Scopus, Web of Science, and Google Scholar databases were obtained using specific keywords.” With “samples, were retrieved from the PubMed, Scopus, Web of Science, and Google Scholar databases using specific keywords.”

Response 1: Thank you for pointing this out. We agree with this comment. Therefore, we have been made the change as noted in the revised manuscript page 1, Lines 14-15.

Comments 2: L 18-20. Replace “The pooled prevalence of E. coli O157:H7 was 4.7%. The pooled prevalence of E. coli O157:H7 from animal samples was 5.4% followed by isolates from environmental samples, 3.4% and 2.8% from isolates from human samples.” With “The pooled prevalence of E. coli O157:H7 was 4.7%, with the highest prevalence observed among animal samples (5.4%) followed by environmental, and human samples (3.4 and 2.8%, respectively).”

Response 2: Thank you for pointing this out. We agree with this comment. Therefore, we have been made the change as noted in the revised manuscript page 1, Lines 19-21.

Comments 3:  Line 29. Replace “Escherichia coli” With “Escherichia coli (E. coli)”

Response 3: We agree with this comment. Therefore, we have been made the change as noted in the revised manuscript page 1, Line 31

Comments 4: L35. Replace “leading” With “leading to”

Response 4: We agree with this comment. Therefore, we have been made the change as noted in the revised manuscript page 1, Line 38

Comments 5: L 38-39. Replace “In Africa, Shiga toxin-producing Escherichia coli (STEC) infections amount to approximately” With “In Africa, the incidence of Shiga toxin-producing E. coli (STEC) infections is approximately”

Response 5: We agree with this comment. Therefore, we have been made the change as noted in the revised manuscript page 1, Lines 40-41

Comments 6:   L 40-41. Replace “contributes 10% to this burden” With “accounts for 10% of this burden.”

Response 6: We agree with this comment. Therefore, we have been made the change as noted in the revised manuscript page 1, Line 43

Comments 7:   L 46. Replace “in; cattle, sheep, goats, beef and meat products, chicken, dairy products, milk, fruits and vegetables; from”. With “in cattle, sheep, goats, beef, meat products, chicken, dairy products, milk, fruits and vegetables from”

Response 7: We agree with this comment. Therefore, we have been made the change as noted in the revised manuscript page 2, Lines 48-49

Comments 8:  L 51. “E. coli” The name of bacteria should be italic, apply this comment throughout the manuscript.

Response 8: We agree with this comment. Therefore, we have been made the change as noted throughout the revised manuscript 

Comments 9: L 51-52.  Replace “E. coli O157:H7 strain is responsible for 20% of E. coli infections in Nigeria [10], and 15.3% in Ethiopia [11], from community-based prevalence studies.” With

“According to the community-based prevalence studies, E. coli O157:H7 strain is responsible for 20, and 15.3% of E. coli infections in Nigeria [10], and Ethiopia [11], respectively.”

Response 9: We agree with this comment. Therefore, we have been made the change as noted in the revised manuscript page 2, Lines 53-55

Comments 10: L 52-54. Replace “In recent years in Africa, several data reported in many systematic reviews have focused on the epidemiology of Escherichia coli using the One Health approach” With

“Recently, several systematic reviews in Africa have focused on the epidemiology of E. coli using the One Health approach”

Response 10: We agree with this comment. Therefore, we have been made the change as noted in the revised manuscript page 2, Lines 55-56

Comments 11:  L 95. Replace “was undertaken and it is the first study that has reported the pooled prevalence” With “was undertaken, for the first time, to report the pooled prevalence

Response 11: We agree with this comment. Therefore, we have been made the change as noted in the revised manuscript page 2, Line 61

Comments 12: L 133. “p” Should be italic, APPLY THIS COMMENT THROUGHOUT THE MANUSCRIPT.

Response 12: We agree with this comment. Therefore, we have been made the change as noted throughout the revised manuscript

Comments 13: L 174, 182, 183, 185, 191. “E. coli”. The name of bacteria should be italic, APPLY THIS COMMENT THROUGHOUT THE MANUSCRIPT.

Response 13:  We agree with this comment. Therefore, we have been made the change as noted throughout the revised manuscript 

Comments 14: 157. “PFGE” Write the full name followed by abbreviation in brackets.

Response 14: We agree with this comment. Therefore, we have been made the change as noted in the  revised manuscript page 5, Line 168

Comments 15: 139. Replace “(Fig.1) (Table 1).” With “(Fig.1 and Table 1)

Response 15: We agree with this comment. Therefore, we have been made the change as noted in the  revised manuscript page 3, Line 132

Comments 16: L184-196. “p” Should be italic, apply this comment throughout the manuscript.

Response 16: We agree with this comment. Therefore, we have been made the change as noted throughout the  revised manuscript

Comments 17: L184. “AMR” Write the full name

Response 17: We agree with this comment. Therefore, we have been made the change as noted in the  revised manuscript page 14, Line 277

Comments 18: L 286. Replace “antimicrobial resistance” With

“antimicrobial resistance (AMR)”

Response 18: We agree with this comment. Therefore, we have been made the change as noted in the  revised manuscript page 14, Lines 279-280

Comments 19: L295. “ESBL” Write the full name followed by abbreviation in brackets.

Response 19: We agree with this comment. Therefore, we have been made the change as noted in the  revised manuscript page 15, Lines 288-289

Comments 20: L296. “blaSHV, blaCTX-M, and blaTEM” The genes’ name should be italic, apply this comment throughout the manuscript.

Response 20: We agree with this comment. Therefore, we have been made the change as noted in the  revised manuscript page 15, Line 290

Comments 21: L 321. Replace “It is worthy of note that imipenem” With “Of note, in the current systematic review imipenem”

Response 21: We agree with this comment. Therefore, we have been made the change as noted in the  revised manuscript page 15, Line 314

Comments 22: L 323. Remove “in this systematic review”

Response 22: We agree with this comment. Therefore, we have been made the change as noted in the  revised manuscript

Comments 23: L 304. Table 2. “AMR” Write the full name.

Response 23: We agree with this comment. Therefore, we have been made the change as noted in the  revised manuscript, table 2, line 333

Comments 24:  L 346-347. Replace “where majority of the publications on E. coli O157:H7 isolated using the One Health perspective can be found” With “with the highest number of publications on E. coli O157:H7 isolated through the One Health approach”

Response 24: We agree with this comment. Therefore, we have been made the change as noted in the  revised manuscript, page 19, lines 339-340

Comments 25: L 350. Replace “An other hand,” With “Additionally,”

Response 25: We agree with this comment. Therefore, we have been made the change as noted in the  revised manuscript, page 19, line 343

Comments 26: L383. Remove “respectively”

Response 26: We agree with this comment. Therefore, we have been made the change as noted in the  revised manuscript, page 19, line 373

Comments 27: L384. Replace “publications.” With “publications, respectively”

Response 27: We agree with this comment. Therefore, we have been made the change as noted in the  revised manuscript, page 19, line 374-375.

Reviewer 3 Report

Comments and Suggestions for Authors   This systematic review of E. coli O157:H7 uses the One Health perspective. This distribution of E. coli highlights the interconnectedness between animals, the environment, and human populations in the transmission and persistence. The review is interesting, and I recommend its publication. I have only few some comments. 1) The figures should be improved. Most of them are difficult to read. Also, please provide more descriptions of the figures that are stand-alone.  2) The presentation of results is very difficult to follow. There is no need to repeat all the tables or figures contents 3) Please ensure that the full names of all abbreviations have been expanded when mentioned for the first time (e.g., ESBL, AMR, etc.). 4) Other minor revision, please see the attached file  

Figure A is not clear and can not be read

Comments on the Quality of English Language

The English could be improved to more clearly express the research

Author Response

Comments 1: The figures should be improved. Most of them are difficult to read. Also, please provide more descriptions of the figures that are stand-alone.

Response 1: The authors appreciate the reviewer's comments. The reviewer’s concern is duly noted. The figures provided in the manuscript were provided following the author's guidelines. Also, detailed descriptions of all images have been provided for all images. According to guidelines, the quality of the figures should not be less than 600 dpi. The quality of all figures in the manuscript is 1200 dpi.

Comments  2: The presentation of results is very difficult to follow. There is no need to repeat all the tables or figures contents

Response 2: The authors appreciate the reviewer's comments. The reviewer’s concern is duly noted. The authors added in details, the important results in the manuscript for the result to be comprehensive. Not all results in the figure or tables were repeated in the text. 

Comments 3: Please ensure that the full names of all abbreviations have been expanded when mentioned for the first time (e.g., ESBL, AMR, etc.).

Response 3: We agree with this comment. Therefore, we have been made the change as noted throughout the revised manuscript.

Comment 4 : Other minor revision, please see the attached file

Response 4 : We agree with this comment. Therefore, we have been made the change as noted throughout the revised manuscript 

Round 2

Reviewer 3 Report

Comments and Suggestions for Authors

Thanks to the authors, the manuscript has improved. However, the figures are of low quality. Maybe the production team will manage. 

Comments on the Quality of English Language

The English could be improved to more clearly express the research.